

# Quantum chaos in perturbative super-Yang-Mills Theory

Tristan McLoughlin*, Raul Pereira† and Anne Spiering‡

School of Mathematics & Hamilton Mathematics Institute, Trinity College Dublin, Ireland

★ tristan@maths.tcd.ie , † raul@maths.tcd.ie , ‡ spiering@maths.tcd.ie

## Abstract

We provide numerical evidence that the perturbative spectrum of anomalous dimensions in maximally supersymmetric $SU(N)$ Yang-Mills theory is chaotic at finite values of $N$. We calculate the probability distribution of one-loop level spacings for subsectors of the theory and show that for large $N$ it is given by the Poisson distribution of integrable models, while at finite values it is the Wigner-Dyson distribution of the Gaussian orthogonal ensemble random matrix theory. We extend these results to two-loop order and to a one-parameter family of deformations. We further study the spectral rigidity for these models and show that it is also well described by random matrix theory. Finally we demonstrate that the finite-$N$ eigenvectors possess properties of chaotic states.

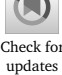

# 1 Introduction

An important sign of chaos in quantum systems is the appearance of random matrix theory (RMT) [1] statistics in the fluctuations of the energy spectrum. For example, the statistics of nearest-neighbour level spacings in integrable systems are generally described by the Poisson distribution [2], while in chaotic systems they are closely approximated by the Wigner-Dyson distribution. This has often been taken as a defining property of chaotic quantum systems and has been seen in a wide variety of areas ranging from condensed matter physics to quantum gravity. In this letter we provide further evidence for the claim, [3], that the perturbative spectra of anomalous dimensions for $\mathcal{N} = 4$ super-Yang-Mills (SYM) with $SU(N)$ gauge group and related theories are well described by the Gaussian orthogonal ensemble (GOE) RMT [4].

If we view gauge-invariant composite operators in $\mathcal{N} = 4$ SYM as analogues of nuclei in QCD, it should not be surprising that their spectrum is described by RMT. After all, it was the use of RMT to describe the statisical properties of large nuclei that inspired many of the initial developments in the study of chaotic quantum systems, first in the work of Wigner [5] and further developed by Dyson, Mehta and many others [6]. The motivation to study the presence of chaos in $\mathcal{N} = 4$ SYM further comes from the fact that it is the canonical example of a holographic theory, dual to type IIB superstring theory, and many of the recent developments in quantum many-body chaos have come from the connection to black hole physics [7–10]. Much of the recent work has focussed on the SYK model of $N$ Majorana fermions with random couplings [11–14], as this model is solvable in the large-$N$ limit despite being strongly chaotic. $\mathcal{N} = 4$ SYM is a much richer and complicated theory, however it is well known to be integrable in the large-$N$ limit where there exist exact results for anomalous dimensions. While at finite values of $N$ integrability is broken, the theory remains superconformal and correlation functions of conformal primary operators remain the natural observables whose properties will be our focus.

# 2 Theory

Quite generally for conformal theories in four-dimensions invariance under the conformal symmetry group $SO(2, 4)$ implies that the two-point functions of primary scalar operators, $\mathcal{O}_i(x)$, can be put in the form

$$\langle \mathcal{O}_i(x_1)\mathcal{O}_j(x_2)\rangle = \frac{\delta_{ij}}{x_{12}^{2\Delta_i}}, \tag{1}$$

with $\Delta_i$ denoting the scaling dimension of operator $\mathcal{O}_i$. The theory is therefore characterized by the scaling dimensions which thus provide basic data regarding the theory. Equivalently, one may focus on the spectrum of the dilatation operator $\mathfrak{D}$, which is the generator of dilatation transformations on composite operators. The $\Delta_i$ which are the eigenvalues of $\mathfrak{D}$ equal the bare dimensions of operators at tree level, but are corrected in the quantum theory.

## 2.1 $\mathcal{N} = 4$ SYM theory

We first focus on the four-dimensional super-Yang-Mills theory with the maximal amount of supersymmetry and our discussion will follow the treatments in [15, 16]. We consider a particular class of local gauge-invariant operators which are given as products of traces of the fields e.g.

$$\mathrm{Tr}(\chi_{1,1}\ldots\chi_{1,L_1})\ldots\mathrm{Tr}(\chi_{m,1}\ldots\chi_{m,L_m})(x), \tag{2}$$

where each field $\chi$ is either a scalar, fermion or field strength with possible insertions of covariant derivatives $\mathcal{D}_\mu$. The fields $\chi$ all transform in the adjoint representation of the gauge

group, which we take to be SU($N$), and so can be expanded in a basis of the gauge algebra generators $T^a$: $\chi = \chi^a T^a$.

The symmetry algebra of the theory, $\mathfrak{psu}(2,2|4)$, is non-compact and thus the operators organize themselves in infinite-dimensional representations. Choosing a basis for the fields $\chi_A$, the generators $\mathfrak{J}$ of $\mathfrak{psu}(2,2|4)$ transform the fields among themselves

$$\mathfrak{J}\chi_A = (\mathfrak{J})^B{}_A \chi_B \, , \tag{3}$$

and in the classical limit, composite operators such as (2) transform in standard tensor product representations. It is convenient to introduce a variational notation for the fields [17]

$$\check{\chi}^A = \frac{\delta}{\delta\chi_A} = T^a \frac{\delta}{\delta\chi_A} \, , \tag{4}$$

so that the action of symmetry generators can be written as

$$\mathfrak{J} = (\mathfrak{J})^B{}_A \mathrm{Tr}(\chi_B \check{\chi}^A) \, . \tag{5}$$

The generators in general receive quantum corrections which can be computed in perturbation theory providing an expression for the generators as a series in the gauge theory coupling constant $g$

$$\mathfrak{J} = \sum_{n=0}^{\infty} g^n \mathfrak{J}_n \, . \tag{6}$$

Of particular importance is the dilatation generator, $\mathfrak{D}$, which at tree-level acts as

$$\mathfrak{D}_0 = \sum_A \dim(\chi_A) \mathrm{Tr}(\chi_A \check{\chi}^A) \, , \tag{7}$$

where $\dim(\chi_A)$ is the classical dimension of the fields. Beyond tree-level the spectrum of dimensions of operators generally receive non-trivial corrections, however operators containing only a single complex scalar

$$\mathrm{Tr}(Z^L)(x) \, , \tag{8}$$

play an important role as they are half-BPS and due to supersymmetry their dimensions receive no quantum corrections. In the large-$N$ limit the symmetry of the theory is further enhanced, making the spectral problem integrable. The spectrum is determined by that of the single-trace operators, which can be viewed as periodic spin chains, and the anomalous-dimension mixing matrix becomes a spin-chain Hamiltonian [18]. In this picture the half-BPS operators correspond to the spin-chain vacua while operators with additional fields correspond to excited states.

We will henceforth restrict to rank-one sectors of the spin chain where there is only a single type of excitation. This will considerably simplify the analysis, but still showcase the universal chaotic behaviour of the theory at finite $N$. First, we consider the $\mathfrak{su}(2)$ scalar sector, with the spin-chain vacuum set by the half-BPS operator in (8) and excited sites given by complex fields $X$. The action of the dilatation operator up to two loops is given by [17,19]

$$\begin{aligned}
\mathfrak{D} = {} & :\mathrm{Tr}(Z\check{Z}): + :\mathrm{Tr}(X\check{X}): -\frac{2g^2}{N}:\mathrm{Tr}([X,Z][\check{X},\check{Z}]): \\
& -\frac{2g^4}{N^2}:\mathrm{Tr}([[X,Z],\check{Z}][[\check{X},\check{Z}],Z]): -\frac{2g^4}{N^2}:\mathrm{Tr}([[X,Z],\check{X}][[\check{X},\check{Z}],X]): \\
& -\frac{2g^4}{N^2}:\mathrm{Tr}([[X,Z],T^a][[\check{X},\check{Z}],T^a]): ,
\end{aligned} \tag{9}$$

where :: denotes that the variational derivatives are normal ordered.

When investigating spectral statistics, it is important to only consider states which share the same global quantum numbers, since different sectors cannot mix. Thus we will focus on sectors with fixed number of fields $L$ and excitation number $M$. Furthermore, as there is a global $SU(2)$ symmetry in this sector, we desymmetrise by restricting to primary operators defined by $J_-\mathcal{O} = 0$, where the lowering operator acts as $J_- X = Z$.

The second sector we examine in this work is the $\mathfrak{sl}(2)$ sector, where a spin-chain site becomes excited by the insertion of covariant light-cone derivatives. These operators are of particular interest as they are in some sense universal in non-Abelian gauge theory [20], and can even be related to perturbative QCD [21] where there is a one-loop integrable $\mathfrak{sl}(2)$ sector at large-$N$ [22, 23]. We denote a site excited with $n$ derivatives by $Z^{(n)} \equiv \mathcal{D}^n Z/n!$. The dilatation operator in this sector, here to just one-loop, is

$$\mathfrak{D}|_{\mathcal{O}(g^2)} = -\frac{g^2}{N} \sum_{\substack{m,n \\ k+l=m+n}} C_{mn}^{kl} : \mathrm{Tr}\big([Z^{(k)}, \check{Z}^{(m)}][Z^{(l)}, \check{Z}^{(n)}]\big) : , \tag{10}$$

with coefficients

$$C_{mn}^{kl} = \delta_{k=m}\big(h(m) + h(n)\big) - \frac{\delta_{k \neq m}}{|k-m|}, \tag{11}$$

with $h(n)$ the harmonic sum. We organize the operators with respect to their number of fields $L$ and derivatives $S$. Furthermore, since there is an $SL(2)$ symmetry, we only consider the operators obeying $S_-\mathcal{O} = 0$, where the action of the lowering operator is given by $S_- Z^{(n)} = n Z^{(n-1)}$.

Finally, the dilatation operator in both sectors is invariant under a parity transformation, $\mathcal{P}$, which reverses the order of fields within a trace

$$\mathcal{P}\,\mathrm{Tr}(\chi_1 \ldots \chi_L)(x) = \mathrm{Tr}(\chi_L \ldots \chi_1)(x). \tag{12}$$

Therefore, to complete the desymmetrization of the mixing matrix, we consider operators with definite parity.

## 2.2 $\beta$-deformed theory

We will also consider $\beta$-deformed $\mathcal{N} = 4$ SYM theory which is an exactly marginal deformation of $\mathcal{N} = 4$ SYM obtained by a modification of its superpotential [24]. This modification introduces a deformed commutator $[.,.]_\beta$ depending on the charges of the fields, and breaks the supersymmetry of the theory down to $\mathcal{N} = 1$. In the $\mathfrak{su}(2)$ sector this deformed commutator is given by

$$[X, Z]_\beta = e^{i\beta} XZ - e^{-i\beta} ZX . \tag{13}$$

For the $SU(N)$ gauge group the $\beta$-deformation preserves the quantum conformal invariance, and the one-loop dilatation operator in the $\mathfrak{su}(2)$ sector is [25], see also [3],

$$\mathfrak{D}|_{\mathcal{O}(g_\beta^2)} = -\frac{2g_\beta^2}{N}\Big( : \mathrm{Tr}\big([X,Z]_\beta[\check{X},\check{Z}]_\beta\big) : + \frac{4}{N}\sin^2(\beta) : \mathrm{Tr}(XZ)\mathrm{Tr}(\check{X}\check{Z}) : \Big) , \tag{14}$$

where $g_\beta$ is the deformed coupling given by $g^2 = |g_\beta|^2(1 - |e^{i\beta} - e^{-i\beta}|^2/N^2)$. It differs from the dilatation operator in the undeformed theory by a replacement of the usual commutators by their deformed analogues and a double-trace correction. At the planar level, the latter is crucial for the vanishing of the anomalous dimension of $\mathrm{Tr}(XZ)$, while for longer operators it leads to a non-planar contribution. For $\beta \in \mathbb{R}$ the dilatation operator reduces to an integrable spin-chain Hamiltonian [26, 27] in the planar limit (and $L > 2$), thus preserving the planar integrability

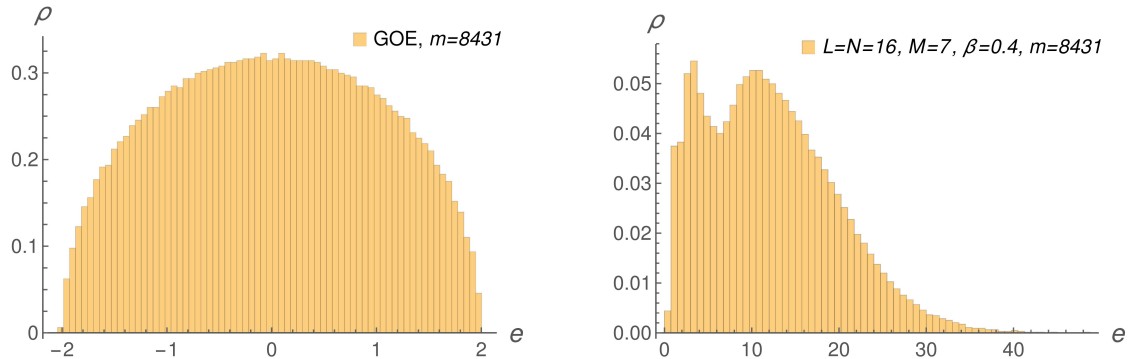

Figure 1: Energy densities for GOE (left) and the $\beta$-deformed theory (right). The number of states $m$ is the same in both cases. The distributions are clearly of a different nature, but we find that fluctuations have a similar behaviour.

of the undeformed theory. In this deformed theory the excitation number is still a conserved quantity, however this $U(1)$ symmetry is no longer part of a larger spin $SU(2)$ symmetry. Thus, sectors of fixed $L$ and $M$ do not further decompose into primary and descendant states which, as we will see, allows for better statistics than in the undeformed case.

In this work we study the behaviour of the spectrum at finite values of the gauge group rank. For $N \geq L$ the dependence of the spectrum on $N$ is due solely to its appearance in the matrix elements of the dilatation operator. However, when $N < L$ there are also relations between single- and multi-trace operators which effectively reduce the size of the Hilbert space. Looking for example at the extreme case of the $SU(2)$ gauge theory, the only surviving states are those built from length-2 traces

$$\mathrm{Tr}(ZZ)^{\frac{L-M-n}{2}} \, \mathrm{Tr}(ZX)^n \, \mathrm{Tr}(XX)^{\frac{M-n}{2}} \,. \tag{15}$$

We compute all such identities when analysing the energy fluctuations for theories with $2 < N < L$. However, for given $L$ and $M$ the spectral statistics become poorer for small $N$ due to the shrinking of the basis of states. It is interesting to note that the dilatation operator in the $N = 2$ case is particularly simple, and the mixing problem becomes solvable in the $\mathfrak{su}(2)$ sector at one loop, yielding the spectrum of energies

$$e_n = 4\cos^2 \beta \, (L + 1 - 2n)n \,, \tag{16}$$

with $n = 0, \ldots, \lfloor M/2 \rfloor$.

## 3 Spacings

The overall energy dependence of the spectral density is usually specific to a theory, see e.g. Figure 1. The Bohigas-Giannoni-Schmit [28] conjecture states that level fluctuations about this overall trend for chaotic quantum systems have universal features described by RMT.[1] To remove the overall trend, we define the cumulative level number

$$n(e) = \sum_{i=1}^{m} \Theta(e - e_i) \,, \tag{17}$$

---

[1]The BGS conjecture was originally for time-reversal symmetric systems whose classical limits are chaotic K systems but is now often taken to hold for general quantum systems.

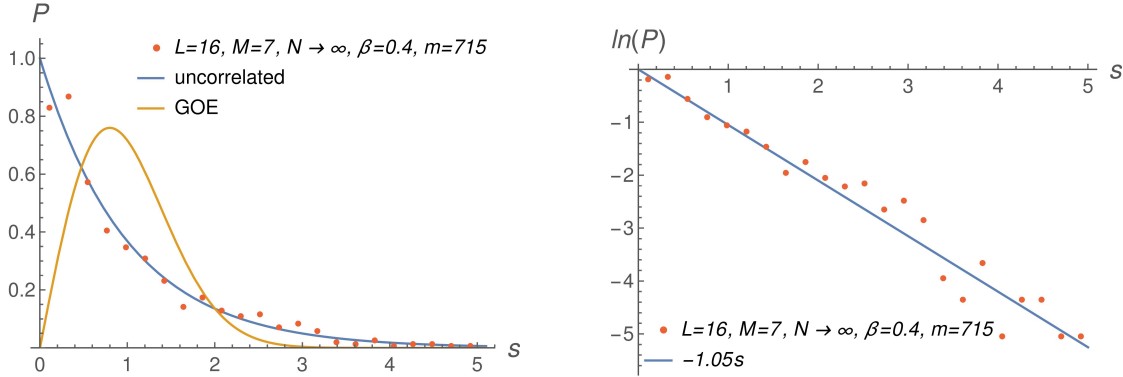

Figure 2: On the left we plot the spacing distribution for the planar $\beta$-deformed theory against the expected Poisson distribution of integrable systems. On the right we linearize the distribution and find that the best fit yields a slope equal to 1.05.

with $m$ the total number of states, and decompose it into an average and fluctuation part

$$n(e) = n_{\text{av}}(e) + n_{\text{fl}}(e). \tag{18}$$

The unfolded spectrum is then given by

$$\varepsilon_i = n_{\text{av}}(e_i), \tag{19}$$

and captures the physics of the spectral fluctuations. As the chaotic features of the fluctuations generally reside away from the edges of the spectrum [29, 30], we perform a polynomial fit with some clipping of low- and high-energy states to find $n_{\text{av}}$, see details in the appendix.

In the large-$N$ limit, the theories under consideration exhibit enhanced symmetries, which lead to their integrable and degenerate spectra. Typically these systems are described by uncorrelated energy levels, and in order to study this behaviour we look at the distribution of spacings $s_i = \varepsilon_{i+1} - \varepsilon_i$, with $\{\varepsilon_i\}$ the ordered unfolded energy levels. The unfolding normalizes the average spacing to one, and so we find that the probability distribution $P$ of spacings is well described by the Poisson distribution

$$P_{\text{P}}(s) = \exp(-s). \tag{20}$$

as can be seen in Figure 2 where we plot the distribution of spacings for $L = 16$, $M = 7$ operators in the $\beta$-deformed theory.

As we make $N$ finite, the theory starts to resemble a chaotic system and we encounter level repulsion, i.e. the probability of coinciding neighbouring levels $P(s \to 0)$ approaches zero. More precisely, we find that the fluctuations in the spectrum behave like those of random matrices, with the spacing distribution well approximated by the Wigner-Dyson distribution

$$P_{\text{WD}}(s) \propto s^\alpha \exp(-A(\alpha)s^2), \qquad A(\alpha) = \frac{\Gamma(1 + \alpha/2)^2}{\Gamma((1 + \alpha)/2)^2}, \tag{21}$$

see Figure 3.

In order to get an estimate for the parameter $\alpha$, we compute the spacings of the unfolded data, bin them and compute the fraction occuring in each of the bins. In the appendix we describe this procedure in more detail and show that the choice of the degree $p$ of the polynomial unfolding, the number of bins $n_b$, and the clipping fractions $f$ does not considerably affect the results. Taking a range of values for $n_b$, $p$ and $f$, we find that the average and

standard deviation values for $\alpha$ in the rank-one sectors of $\mathcal{N} = 4$ SYM theory we considered are

$$\alpha_{\mathfrak{su}(2)} = 1.01 \pm 0.01\,,$$
$$\alpha_{\mathfrak{sl}(2)} = 0.99 \pm 0.01\,. \tag{22}$$

The first corresponds to the $\mathfrak{su}(2)$ sector with $L = 17$ and $M = 6$, while the second is for the $\mathfrak{sl}(2)$ sector with $L = 11$ and $S = 7$. This shows that the fluctuations in both sectors of the theory are well described by GOE RMT. This is expected as the action of the dilatation operator is symmetric under a time-reversal transformation.

If we include the effects of the two-loop dilatation operator, the qualitative behaviour is unchanged as can be seen in Figure 4 (left). This is in part a consequence of the fact that the spectrum of the two-loop part of the dilatation operator, by itself, has a GOE Wigner-Dyson distribution Figure 4 (right) with $\alpha = 0.89$. It is also the case that the planar limit of the two-loop dilatation is not exactly integrable but only integrable up to $\mathcal{O}(g^4)$, which can be seen in the planar spectrum itself not being Poissonian for finite $g$. We should make clear that, as the perturbative expansion is asymptotic at finite-$N$, we are not making conclusions about the finite-$g$ behaviour of the spectrum, but rather are only exploring the qualitative effect on the perturbative spectrum of including higher-order terms in the dilatation operator.

Meanwhile, looking at the $\beta$-deformed theory in the $L = 16$ sector, we find that

$$\alpha_{N=16} = 0.93 \pm 0.01\,,$$
$$\alpha_{N=4} = 0.74 \pm 0.01\,, \tag{23}$$

showing that the GOE distribution is still a good approximation as we decrease the value of $N$, see Figure 5. The fit for $N = 4$ is clearly worse, but that could be due to the poorer statistics inherent to the smaller size of the Hilbert space, which is reduced due to trace identities.

## 4 Spectral Rigidity

Another feature of the level fluctuations in chaotic systems is spectral rigidity which is measured by the Dyson-Mehta statistic $\Delta_3$. While the nearest-neighbour spacing distribution measures short-range correlations in the spectrum, this quantity provides information about long-range correlations and is related to the variance of the level number. It is a function of an interval length $l$ [31]

$$\Delta_3(l) = \frac{1}{l} \left\langle \min_{A,B} \int_{\varepsilon_0}^{\varepsilon_0 + l} d\varepsilon\, (\hat{n}(\varepsilon) - A\varepsilon - B)^2 \right\rangle\,, \tag{24}$$

with

$$\hat{n}(\varepsilon) = \sum_{i=1}^{m} \Theta(\varepsilon - \varepsilon_i) \tag{25}$$

the unfolded cumulative level number. The expression inside the angle-bracket computes the least-square deviation of $\hat{n}(\varepsilon)$ from the best straight line fitting it in the interval $[\varepsilon_0, \varepsilon_0 + l]$, while the bracket $\langle .. \rangle$ denotes an average over values of $\varepsilon_0$ taken from a discretization of the interval $[\varepsilon_1, \varepsilon_m - l]$. For a given integration window with $k$ levels inside, it is useful to parametrize the levels by

$$\varepsilon_0 + l\, z_i\,, \tag{26}$$

with $0 < z_1 < \ldots < z_k < 1$, so that the expression in (24) simplifies to

$$\Delta_3(l) = \left\langle 6 s_1 s_2 - 4 s_1^2 - 3 s_2^2 + s_3 \right\rangle\,, \tag{27}$$

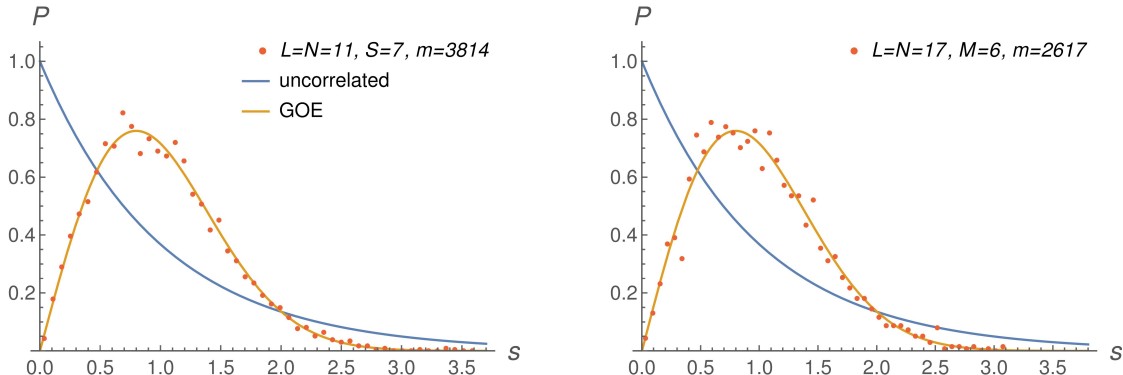

Figure 3: Distribution of spacings for different sectors of $\mathcal{N} = 4$ SYM at $N = L$, against the GOE distribution. On the left we consider the $\mathfrak{sl}(2)$ sector, and on the right we have the $\mathfrak{su}(2)$ sector. In both cases we focus on the positive parity sector, and clip 11% and 4% of low- and high-energy states respectively.

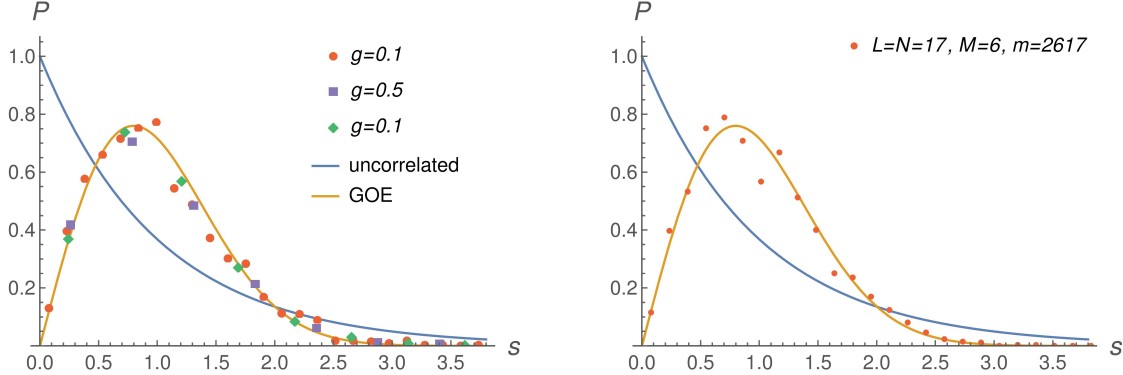

Figure 4: Distribution of spacings for the dilatation operator through two-loops for $g = 0.1, 0.5, 1$ (left) and for the two-loop part of the dilation operator by itself (right) in the $\mathfrak{su}(2)$ sector. In both cases we focus on the positive parity sector of $L = 17$ states, and clip 11% and 4% of low- and high-energy states respectively.

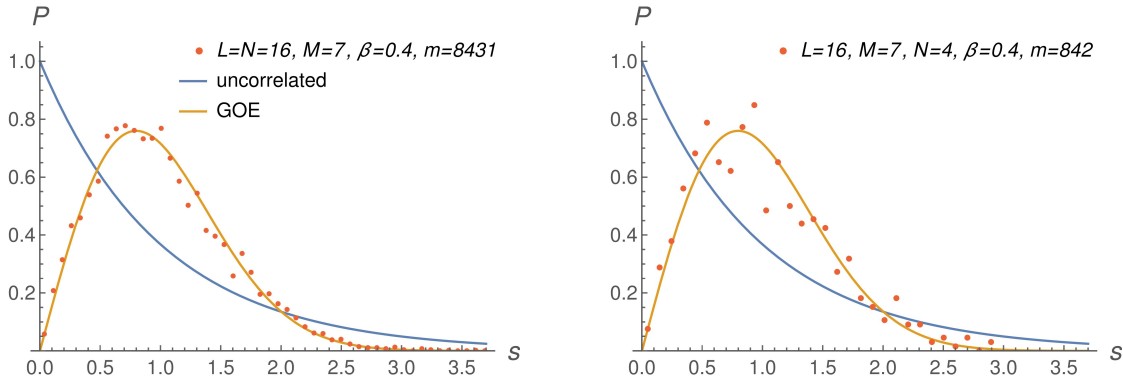

Figure 5: Distribution of spacings (in red) for the $\beta$-deformed theory at different values of $N$, against the GOE distribution (in blue). On the left we consider $N = 16$, while on the right we focus on $N = 4$. We clipped 8% (4%) of low-(high-)energy states.

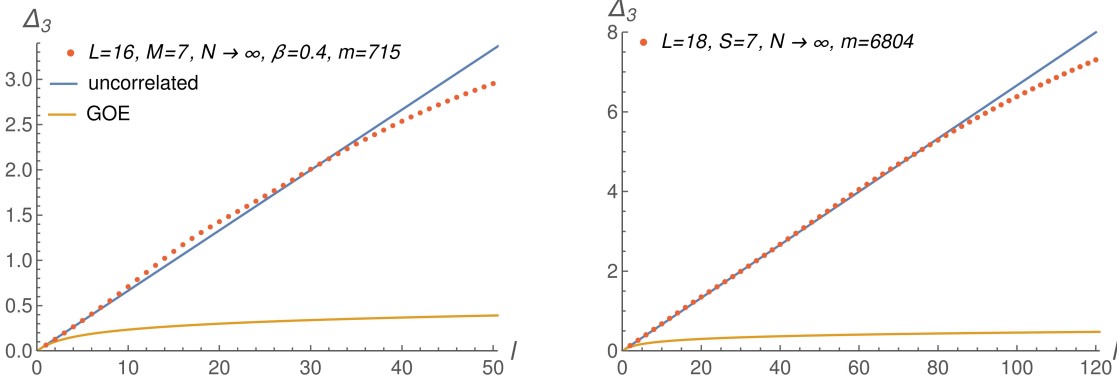

Figure 6: On the left we plot $\Delta_3$ for the planar $\beta$-deformed theory in the $\mathfrak{su}(2)$ sector. The single-trace sector contains only 715 states, which could justify the deviation to the expected behaviour. On the right we plot $\Delta_3$ in the $\mathfrak{sl}(2)$ sector of planar $\mathcal{N} = 4$ SYM, where integrability allows us to find the spectrum exactly. There we have 6804 states and the result matches the linear behaviour all the way up to $l \approx 80$.

where the $s_i$ are defined by the sums [32]

$$s_1 = \sum_{i=1}^{k} z_i \,, \quad s_2 = \sum_{i=1}^{k} z_i^2 \,, \quad s_3 = \sum_{i=1}^{k} (2k - 2i + 1) z_i \,. \tag{28}$$

This allows us to evaluate $\Delta_3$ efficiently for large values of $l$, where larger values of $k$ contribute.

For uncorrelated fluctuations $\Delta_3$ grows linearly in $l$, specifically for the Poisson distribution

$$\Delta_3(l) = \frac{l}{15} \,. \tag{29}$$

In Figure 6 we show that the planar spectra of both the $\mathfrak{su}(2)$ sector of $\beta$-deformed $\mathcal{N} = 4$ SYM and the $\mathfrak{sl}(2)$ sector of $\mathcal{N} = 4$ SYM follow (29) up to some non-universal length $l_{max}$. This behaviour of $\Delta_3$ is characteristic of integrable models and was proved for semi-classical integrable models in [33].

The behaviour of $\Delta_3$ for a GOE RMT at small $l$ corresponds to that of the Poisson distribution (29), while for large $l$ one finds the slower growth [1]

$$\Delta_3(l) = \frac{1}{\pi^2} \left( \ln(2\pi l) + \gamma_E - \frac{5}{4} - \frac{\pi^2}{8} \right) + \mathcal{O}(l^{-1}) \,. \tag{30}$$

In [33] it was also shown that certain semi-classical chaotic systems follow this behaviour. For non-planar Yang-Mills theories, we find that $\Delta_3$ clearly grows slower than the integrable case for large lengths and follows (30) for large lengths up to some non-universal $l_{max}$, as demonstrated in Figure 7[2] However, we see that $\Delta_3$ of the $\beta$-deformed theory at $N = 4$ does not match the GOE prediction quite as well. This could be a result of the smaller Hilbert space, but also a small $N$ effect as we approach the solvable $SU(2)$ gauge theory.

---

[2]For plotting the GOE prediction we numerically evaluate the expressions given in [1]. We thank J. Verbaarschot for discussions on this and for sharing his Mathematica code.

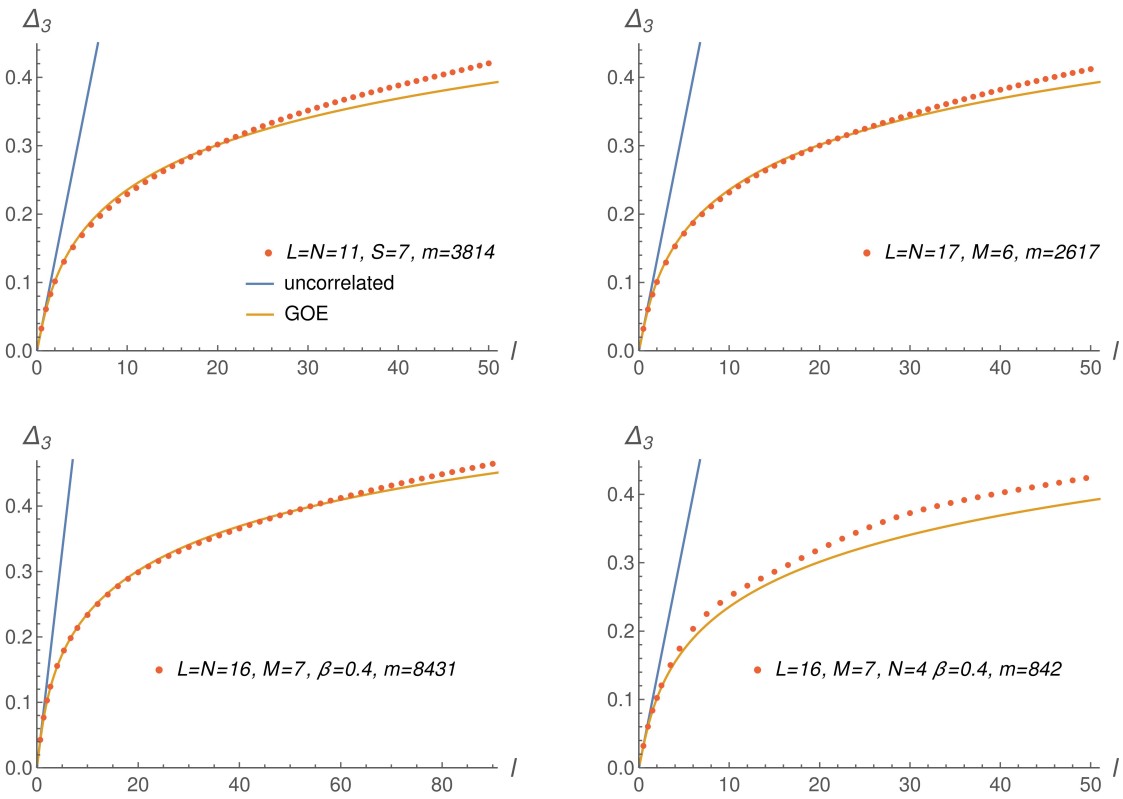

Figure 7: In the top row we depict $\Delta_3$ for the $\mathfrak{sl}(2)$ (left) and $\mathfrak{su}(2)$ (right) sectors of $\mathcal{N}=4$ SYM and observe excellent agreement with the GOE RMT prediction up to $l \approx 25$. In the bottom row we plot $\Delta_3$ for the $\beta$-deformed theory at $N=17$ (left) and $N=4$ (right). The former matches perfectly the expected behaviour up to $l \approx 90$, while the latter is close to, but consistently above, the result of the GOE.

## 5  Chaotic Eigenstates

Going beyond the energy spectrum, it is interesting to study the properties of the eigenstates at different values of $N$. GOE RMT makes a number of predictions for the distribution of chaotic eigenstates, in particular that they are spread out over any non-finely tuned reference basis, i.e. they are delocalised. As a measure of this spreading we use the information entropy for each eigenvector $|e_k\rangle$

$$S_k = -\sum_{a=1}^{m} |c_{ka}|^2 \ln |c_{ka}|^2, \tag{31}$$

where the coefficients $c_{ka}$ are taken with respect to the reference basis $|a\rangle$: $|e_k\rangle = \sum_{a=1}^{m} c_{ka}|a\rangle$. As our choice of reference basis we simply take the multi-trace operators with fixed numbers of excitations which was used to compute the dilatation operator matrix elements. The GOE RMT prediction is that $S = \ln(2m) + \gamma_E - 2 + \mathcal{O}(1/m)$ for all eigenvectors see e.g. [29,34]. However in most physical systems, for example in nuclei [29] or spin-1/2 spin chains [35], the RMT result is only approached near the middle of the energy band, while the states at the edges have significantly lower entropy. In Figure 8 we plot the information entropy, normalised to the corresponding RMT values, for the $\beta$-deformed theory for a range of values of $N$. It is clear that the entropy is larger for smaller values of $N$ and the mean entropy at $N = L$ is 0.94, which is significantly larger than the value at large $N$, 0.38. Perhaps even more noticeably, the

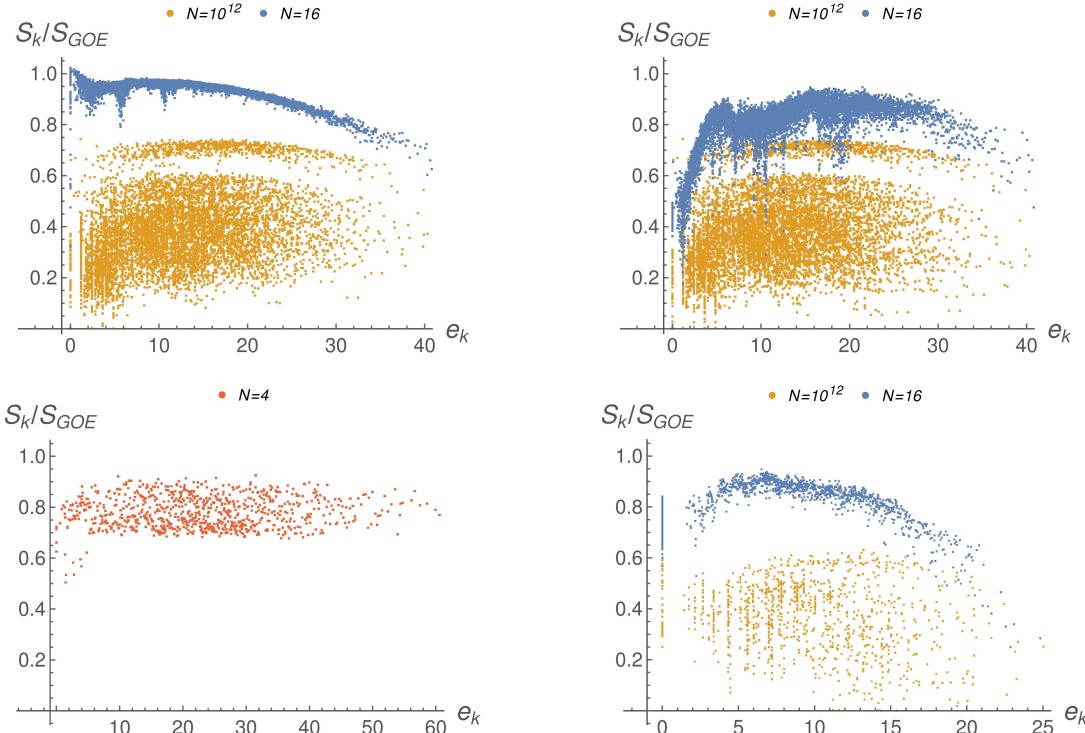

Figure 8: Information entropy of eigenvectors from the $L = 16$, $M = 7$ sector of the $\beta$-deformed theory at $N = 10^{12}$ (top left, yellow, $m = 8512$), $N = 16$ (top left, blue, $m = 8512$) and $N = 4$ (bottom left, $m = 846$). On the top right we consider the same quantities but for the transpose of the mixing matrix. The information entropy of the two-loop dilatation operator of the undeformed theory in the positive-parity sector is plotted for both $N = 10^{12}$ (bottom right, yellow, $m = 1201$) and $N = 16$ (bottom right, blue, $m = 1201$).

fluctuations in the entropy values are much smaller at $N = L$ where $S_k$ somewhat resembles a smooth function of the energy.

One issue in comparing RMT with the gauge theory is that while the dilatation operator has real eigenvalues and possesses discrete symmetries analogous to time-reversal, with the choice of a scalar product for which the basis of multi-trace operators is orthonormal, it is in fact not symmetric. As a consequence, its eigenvectors are complex rather than real and are not mutually orthogonal with respect to our scalar product. This can be seen in the different entropies of the eigenvectors of the transposed matrix, Figure 8, though one still notices that the finite-$N$ states generally still have larger entropy.

We also plot the case of $L = N^2$, and again one finds that the mean value is well above the integrable large-$N$ result, and the maximum value 0.92 approaches the RMT bound. As the dimension of the $N < L$ Hilbert space is smaller, the statistics are perhaps not as reliable, but one interesting feature is the uniformity of the entropy with mixing being almost independent of the energy. The information entropy can be similarly computed for the two-loop dilatation operator in the undeformed theory. In Figure 8 we consider the value $g = 0.1$, and while there is a number of differences in the structure of the states, qualitatively the results are similar. We can repeat these calculations for the $\mathfrak{sl}(2)$ sector and while we find that the mean entropy is still larger at small $N$ than large $N$, it is generally quite low and significantly further from the GOE RMT bound.

# 6 Conclusion

We have provided numerical evidence that the non-planar spectrum of $\mathcal{N} = 4$ SYM, in spite of its maximal supersymmetry, exact conformal invariance and planar integrability, is quite generic and shares the universal properties we expect of chaotic quantum many-body systems. While we have only displayed results for specific choices of operator length and excitation number, we have found comparable results for all other charges that we have considered. While it is possible that by resumming the perturbative series and including non-perturbative effects the qualitative behaviour will change, it is natural to conjecture that the non-planar spectrum is described by GOE RMT at finite values of both $g$ and $N$. Such a conjecture is further motivated by the fact that, at strong coupling $g \gg 1$, operators with dimensions $\Delta \sim N^2$ are holographically dual to Black Holes and so are expected to exhibit chaotic properties [7–10].

This connection was pursued recently in [36], where the authors considered operators dual to a system of giant gravitons. In that sector the one-loop dilatation operator can be reformulated as a Hamiltonian acting on a graph [37], with a large-$N$ counting of graphs matching the expected black hole entropy and therefore suggesting an interpretation of the operators as black hole microstates. The natural basis in this context is that of restricted Schur polynomials, which diagonalize the free-theory two-point functions at finite $N$. It could be particularly interesting to study the energy eigenfunctions in this basis, where the dilatation operator becomes symmetric, and find the implications for the information entropy.

While strong coupling can be difficult to access from the gauge theory, an interesting connection with gravity has been made in the context of the SYK model, which saturates the bound on chaos [38]. In this model, the behaviour of fluctuations is also controlled by random matrix dynamics, with the relevant Gaussian ensemble determined by the number of fermions in the model [39]. RMT was also shown to describe well the late-time behaviour of the spectral form factor which encodes correlations between separated energy levels [40].

More generally, the chaotic behaviour of classical Yang-Mills has been long studied, see [41, 42] and [43] for a review of early work.[3] In particular it was shown that upon restriction to homogeneous field configurations, and similarly in the reduction of the classical theory on $\mathbb{R} \times S^3$, the resulting Hamiltonian is non-integrable. The dilatation generator for $\mathcal{N} = 4$ SYM on $\mathbb{R}^{1,3}$ studied in this work is the Hamiltonian for the theory on $\mathbb{R} \times S^3$ and the corresponding reduction to one dimension is the BMN matrix theory [44] which is known to be classicaly chaotic [45]. It is also known that the one-loop anomalous dimensions of the matrix theory agree with the full theory, though this agreement does not persist at two-loops, [46] while the planar theory is integrable at least up to fourth order [47]. It would be reasonable to further conjecture that at finite-$N$ this theory will similarly have a spectrum described by GOE RMT.

It is interesting to note that GOE statistics have previously appeared in the related context of strings moving in space-times holographically dual to confining theories [48]. In this case the chaotic behaviour is not a finite-$N$ effect but is rather due to the non-integrability of the string worldsheet [49] which also occurs in many other holographic backgrounds e.g. [50–53]. It would be interesting to extend the analysis of spectra carried out in this work to gauge theories with such holographic duals as it may give a deeper understanding of the role of chaos in this context. RMT dynamics have also been observed is the D1-D5 CFT. This system was studied at weak-coupling in [54] which demonstrated that, by using a progressive time averaging, the two-point function had a characteristic decay-ramp-plateau behaviour.

---

[3]We are grateful to A. Tseytlin for bringing these works to our attention.

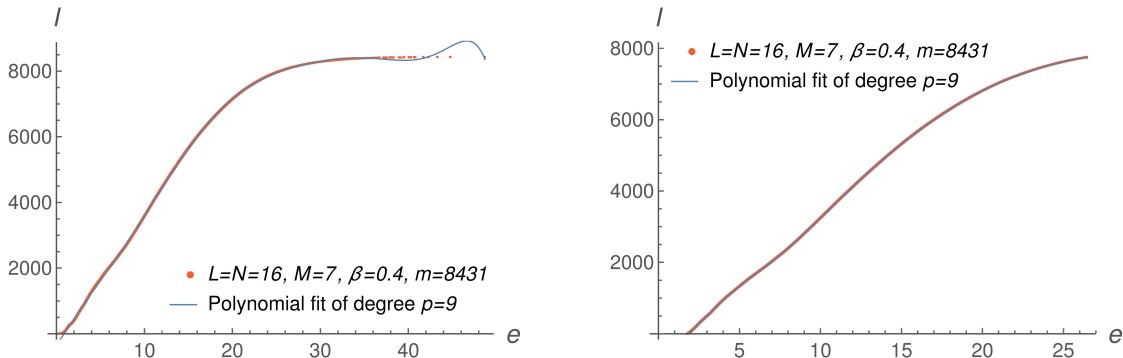

Figure 9: Plot of $n(e)$ for the $\beta$-deformed theory and a polynomial fit of degree 9. On the left we fit the whole spectrum, but obtain poor behaviour for extreme values of $e$. On the right we clip 4% of the states at each end, and find that the fit captures $n_{\mathrm{av}}$ more accurately.

# Acknowledgements

We would like to thank Andrew Cleary and Taillte May for their contributions to undergraduate summer research projects which overlapped with initial parts of this work. We also thank Leopoldo Pando Zayas, Arkady Tseytlin, and Jacobus Verbaarschot for useful comments. Some of the calculations reported here were performed on the Lonsdale cluster maintained by the Trinity Centre for High Performance Computing. This cluster was funded through grants from Science Foundation Ireland.

**Funding information** This work was supported by the Science Foundation Ireland through grant 15/CDA/3472 and has received funding from the European Union's Horizon 2020 research and innovation programme under the Marie Skłodowska-Curie grant agreement No. 764850 "SAGEX".

# A    Data Preparation

In order to find $n_{\mathrm{av}}$, one can select each $n$-th energy state and perform a piecewise linear interpolation. However, we find that with this procedure some of the results change considerably as one varies $n$. Therefore, we choose to approximate $n_{\mathrm{av}}$ with a polynomial fit to the set of $\{n(e_i)\}$. The degree $p$ of the polynomial is a parameter that needs to be tuned, but we find that the results are virtually unchanged for a wide range of values.

Since $n(e)$ is usually quite flat at the ends of the spectrum, we find it furthermore useful to clip at least 4% of the states on each end before fitting the distribution, see Figure 9. While this clipping is necessary in order to obtain a good unfolding, there are still states at the ends of the spectrum that do not exhibit the chaotic properties of RMT. The fraction, $f$, of such states to be removed is theory-dependent, as seen in Figure 10, but can be found systematically. Regarding the degree $p$ of the polynomial unfolding and the number of bins $n_b$, the variations are small, see Figure 11. We use $p = 17$ in all examples shown in this work.

For $\Delta_3$ it is also important to remove a sufficient number of low-energy states. In Figure 12 we plot the ratio $r$ of $\Delta_3(30)$ for a given theory with that of the GOE. One can see that the ratio converges as one increases the clipping fraction. The stabilization in the case of $\Delta_3$ seems to occur at slightly higher values of $f$ than those found in the analogous analysis of the spacings in Figure 10, but is otherwise very similar. Meanwhile, the dependence on the degree of the polynomial unfolding is negligible for $p \geq 11$ and also consistent with the plot in Figure 11.

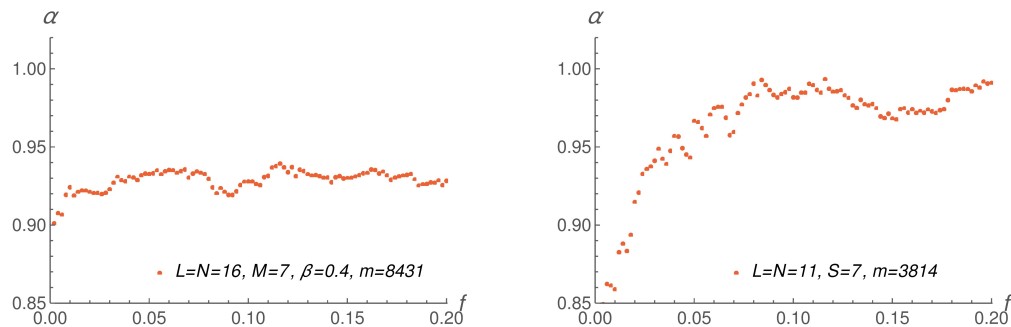

Figure 10: Values of $\alpha$ for the Wigner-Dyson distribution which best fit the data under different clipping fractions $f$ of low-energy states (we clip 4% of the high-energy levels). On the left we consider the $\beta$-deformed theory, while on the right we consider the $\mathfrak{sl}(2)$ sector of $\mathcal{N} = 4$, with $n_b = 50$ in both cases. In order for the results of the undeformed theory to be stable, one needs to clip a larger percentage of states.

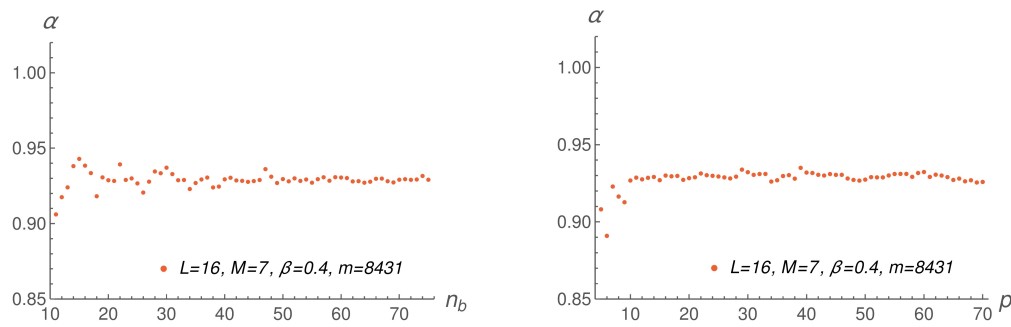

Figure 11: Variation of the best fit parameter $\alpha$ in the $\beta$-deformed theory, with 8% (4%) clipping of low-(high-) energy states. On the left we vary the number of bins $n_b$ used for obtaining the distribution of spacings, with a fixed degree for the polynomial unfolding. On the right we vary the degree $p$ used for the polynomial unfolding of the spectrum, with $n_b = 50$. The results are rather insensitive to variations in these parameters, as long as $p \geq 11$ and $n_b \geq 20$.

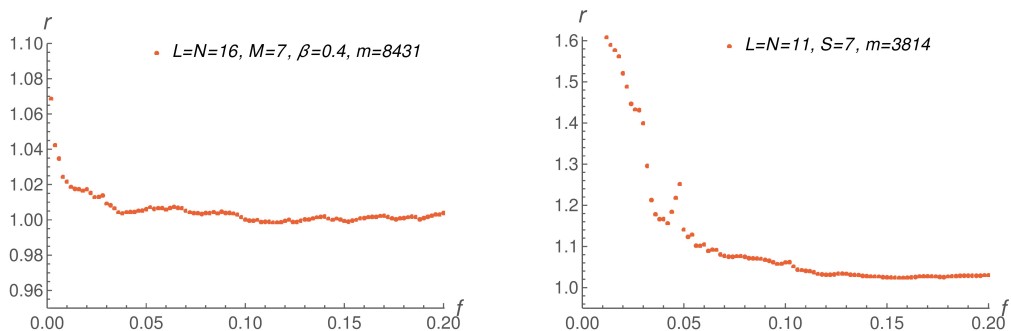

Figure 12: We plot the ratio between $\Delta_3$ of Yang-Mills theories and GOE for $l = 30$, as we vary the clipping fraction. On the left we show the $\mathfrak{su}(2)$ sector of the $\beta$-deformed theory, while on the right we plot the $\mathfrak{sl}(2)$ sector of the undeformed theory. Just as in the context of the spacing distribution, the undeformed theory requires a larger removal of states at the beginning of the spectrum.

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
