# Peer review of "Quantum Chaos in Perturbative super-Yang-Mills Theory"

_SciPost Physics, doi:SciPost Phys. 14, 049 (2023)_

## Round 1 · Referee Report · Robert de Mello Koch (Referee 1) · 2021-5-30

Report

This paper considers well chosen subsectors of ${\cal N}=4$ super Yang Miils theory and it's $\beta$ deformed cousins. In these subsectors the spectrum of the dilatation operator is computed at finite $N$. This entails using a basis of operators that have arbitrary multi-trace structure and taking into account finite $N$ trace relations. The large $N$ spectrum shows Poisson statistics (expected for an integrable system), while the finite $N$ spectrum follows Wigner-Dyson statistics, a sure sign of ergodicity and clear link to random matrix theory. The connection to random matrix theory is further strengthened by studies of spectral rigidity and chaotic eigenstates.

While the spectrum of the planar dilatation operator is well studied, this paper points out a facinating new direction, clearly extending the existing literature in a highly non-trivial way. Indeed, some of the most interesting questions, motivated by holography, are naturally posed for the finite $N$ theory. This paper has developed an interesting set of finite $N$ questions that can be addressed numerically. In addition, the discovered links to random marix theory are compeling and interesting, although not entirely unexpected.

I thoroughly enjoyed reading this paper and am convinced of its value. For this reason I have recommended that it is accepted. I would however encourage the authors to state the dimension of the space that the dilatation operator acts on, in Figure captions. This would be an extremely useful piece of information for others wishing to reproduce the published results.

---

## Round 1 · Referee Report · Chethan Krishnan (Referee 2) · 2021-7-22

Report

The paper sharpens the observation that at finite N, (the dilatation operator of) N=4 SYM is not integrable. The authors approach the problem by explicitly diagonalizing the perturbative dilatation operator, at finite N for subsectors of operators, and noting that the unfolded level spacing distribution is not Poisson, instead it is one of the RMT classes. This is a nice result, and worthy of publication. They also investigate some related RMT features like spectral rigidity etc. The beta deformed N=1 theory is also discussed.

I have some minor comments, which may improve the quality of the paper. But I leave it to the authors whether they decide to make these changes, the paper is publishable as it stands.

Firstly, the paper seems to be written in a letter format. This is fine for the most part, but since this is a paper which is more or less self-contained once one takes the dilatation operators presented in (4), (5) and (9) as a given, I think it may be beneficial to provide an introductory discussion on the dilatation operator. This will make the paper immediately accessible to a much wider audience than the integrability community. The discussion can be in a small appendix, something like page 6 of hep-th/0307015.

Also it may be useful to give some references that orient the present paper in the broader landscape of physics -- From the stringy side as opposed to the gauge theory side, the observation that integrability is lost due to chaos in some N=1 backgrounds (even at infinite N, but on the string worldsheet) has been noted by Basu and Pando Zayas in 1103.4107. Another curious paper that is loosely related is one by Craps et al. 1612.04334, they find that even in the weak coupling limit of the D1-D5 system, at large N, probes behave chaotically. Since the finite N discussions of the present paper can be viewed as a step towards black hole physics, these examples of chaos may be worth pointing out.

---

## Round 2 · Author Response

We thank the referees for their reports and apologise for the long delay in the resubmission. We have attempted to make minor revisions to address the points raised by the referees. In particular:

We have extended the introduction to N=4 SYM in section 2.1 giving some further details about the action of the symmetry generators and the dilatation operator in particular. We have added the references to Basu and Pando-Zayas (and an earlier work by Pando Zayas and Terrero-Escalante) to our discussion chaotic string worldsheet theories. We have added a brief comment on the work by Balasubramanian et al.

Following the second referees suggestion we have added the number of states i.e. the dimension of the Hilbert space to all our figures.

---

## Editorial Decision

published